**Data Availability Statement:** The data is stored at the public repository of the University of Bern: https://boris.unibe.ch/id/eprint/159890.

# Fatigue during the COVID-19 pandemic: Evidence of social distancing adherence from a panel study of young adults in Switzerland

**Axel Franzen**[ID]*[☯], **Fabienne Wöhner**[ID][☯]

Institute of Sociology, University of Bern, Bern, Switzerland

☯ These authors contributed equally to this work.
* axel.franzen@unibe.ch

## Abstract

In this paper we analyze panel data (N = 400) to investigate the change in attitudes towards the Covid-19 measures and the change in compliance behavior between the first and second lockdowns in a sample of young adults from the University of Bern, Switzerland. We find considerable fatigue. While respondents expressed high acceptance of and compliance with the Covid-19 measures during the first lockdown, both acceptance and compliance behavior decreased substantially during the second lockdown. Moreover, we show via a structural equation model that respondents' compliance behavior is largely driven by the perception of how others behave and by the acceptance of the Covid-19 measures. All other effects scrutinized e.g., individual and social risk perception, trust in politics, and pro-social orientations affect compliance behavior via the acceptance of Covid-19 measures. We also conduct two tests of causality of the estimated relation between attitudes towards the measures and social distancing behavior. The first test incorporates the effect of compliance behavior reported during the first lockdown on attitudes during the second lockdown. The second test involves estimating a first difference panel regression model of attitudes on compliance behavior. The results of both tests suggest that the effect of Covid-19 attitudes on social distancing behavior can be interpreted causally.

## 1. Introduction

Switzerland has experienced two lockdowns so far during the Covid-19 pandemic. The first started in the middle of March 2020 and lasted for 6 weeks until the end of April 2020. The second lockdown started on January 18, 2021. The first easing of measures occurred on March 1, 2021, but severe restrictions lasted until April 19, 2021. Generally, the measures during the lockdowns included the closure of shops, restaurants, public facilities, cancellation of events, closure of schools and universities, restrictions in the form of meeting no more than five people, obligations to wear masks, and the recommendation to stay home as much as possible. Additionally, during the second lockdown working from home was mandatory whenever possible. Generally, the measures imposed by the Swiss government were less restrictive as

**Funding:** The author(s) received no specific funding for this work.

**Competing interests:** The authors have declared that no competing interests exist.

compared to other European countries. Thus, there was never any curfew in Switzerland as for instance in Germany, France or Italy.

Since Covid-19 infections cause severe health risks predominantly for the elderly population, compliance with measures designed to avoid the spread of the virus constitutes a social dilemma for the young population. Cooperation in the form of compliance with these measures is good for society but comes along with substantial individual costs. Hence, young people who have a low health risk face incentives not to contribute to the public good through staying at home and limiting social contacts. Hence, in a previous study we analyzed how young adults behaved during the first lockdown [1]. We found that acceptance of and compliance with the measures were surprisingly high during the first lockdown. Most young adolescents believed that the measures were meaningful in an effort to avoid the spread of the virus and they adhered voluntarily even to those measures that were not enforced, such as staying at home and not meeting friends and relatives. In fact, we found that high acceptance of the measures was the most important prerequisite to compliance with the social distancing measures.

The results we present in this article are a continuation of our first study. In particular, we surveyed the same individuals right at the end of the second lockdown asking again to what degree the imposed measures are accepted and to what extent they followed the social distancing restrictions. Our interpretation of the first study was that the unusual emergency situation caused high compliance and high solidarity. One of our main research questions in this second study is to what extent the prolonged nature of the pandemic causes fatigue, and how this fatigue affects acceptance of and compliance with the Covid-19 measures.

The remainder of this article is structured into four sections. Section two presents a short literature overview of what is known so far about compliance in the Covid-19 pandemic. The third section then describes the data and presents some descriptive results on how acceptance of and compliance with the measures changed between the first and the second lockdowns. In section four we present the multivariate results of a structural equation model trying to explain compliance with the Covid-19 measures. Since we are interested in causal explanations of compliance behavior we conduct a number of tests to exclude the possibility that acceptance of the measures is not a cause but rather a rationalization of the behavior. We do this by first investigating to what extent compliance during the first lockdown influences the acceptance of measures during the second lockdown. Furthermore, we conduct a first-difference panel regression model of how the changes in attitudes are related to the changes in behavior. Finally, section five concludes and discusses the results.

## 2. Literature review

There are numerous studies describing and investigating compliance with Covid-19 measures in various countries, particularly for the US [2–7], Germany [8], Italy [9], Switzerland [1, 10], Slovakia [11], Israel [12], South Korea [13], Japan [14], Indonesia [15], or Côte d'Ivoire [16]. Even though many of these studies only use opportunity samples or student samples, a few consistent results have still emerged. Generally, many studies report high compliance with measures against the spread of Covid-19 such as keeping physical distance or engaging in frequent hygiene-related precautions. However, there are also some socio-demographic differences. Thus, women adhere more often to preventive measures than do men [1, 5, 7, 10, 13–15]. Less consistent are the findings concerning age. Most studies report that younger people are less inclined to comply with Covid-19 measures [5, 11], but there is also opposing evidence [2, 4, 14].

More interesting are findings concerning individuals' psychological dispositions like self-control, trust in government and health institutions, and risk perception. Not surprisingly,

subjects perceiving a higher risk of becoming infected exhibit more preventive behaviors (e. g. [7, 8, 11, 13, 14]. There are a number of studies investigating the effect of certain psychological dispositions on non-compliance with coronavirus safety measures. In particular, Nivette et al. [10] found in a sample of young Swiss adults that those with low self-control report less compliance with social distancing measures. Similarly, O'Connell et al. [2] report for a US sample that individuals who report anti-social behaviors (measured by the Subtypes of Antisocial Behavior Questionnaire (STAB)) also comply less with social distancing measures. A large number of psychological dispositions were considered in a study by Bailey et al. [4] using a US convenience sample. They found that those individuals with high behavioral emotional regulation skills and high values of agreeableness—a dimension of the Big Five—show greater compliance with social distancing recommendations.

Many studies also investigate how trust in the government and countries' health institutions affects compliance with social distancing recommendations [11, 12, 16–19]. Pak et al. [17] use a global survey on Covid-19 attitudes and behaviors conducted in 177 countries (see Fetzer et al. [20]). Their findings for the 58 countries included in the final analysis suggest that trust in the government amplifies the following of governmental restrictions implemented in order to avoid Covid-19. The finding that trust increases compliance with governmental restrictions is also supported by national studies for Cote d'Ivoire [16], Israel [12], France [19], and Slovakia [11]. In a similar vein, Farjam et al. [18] demonstrate via a survey experiment that participants react more responsively to Covid-19 measures when the recommendations were given by scientific experts as compared to politicians, leading to the assumption that trust makes a decisive difference.

Summing up, despite the fact that most studies concerned with compliance behavior during the first Covid-19 lockdown use non-random opportunity samples or are limited to specific populations (e.g. students) a few findings appear to be relatively consistent: Individuals seem to comply more often with the coronavirus measures if they are female and older, perceive higher personal risk associated with becoming infected or if they are surrounded by vulnerable relatives or friends. Furthermore, those who have shown antisocial behavior before, or have a certain psychological disposition (e.g. low self-control), also seem less likely to comply with coronavirus measures. Additionally, all studies incorporating trust in governments or health institutions suggest that trust in official representatives increases engagement in social distancing or hygiene measures.

The purpose of this study is to investigate attitudes towards Covid-19 measures and compliance behavior during the second lockdown and to compare this with attitudes and behaviors during the first lockdown. We are particularly interested in how attitudes and behaviors changed between the first and the second lockdown. The main results of our first study were that young adults in Switzerland expressed high support of the Covid-19 measures and complied strictly with social distancing measures. Since young adults have a low risk of severe health problems when infected with Covid-19, complying with measures is like contributing to a public good [1, 21–24]. Keeping social distance and staying at home avoids the spread of Covid-19, and hence, is beneficial for society. But social distancing comes with individual costs, particularly for young adults, and therefore provides an incentive not to comply with social distancing measures. In light of the fact that cooperation is often low in many public goods (e.g. environment, overpopulation) the finding that individuals cooperated during the coronavirus pandemic is surprising. We believe this high level of cooperation was due to the special emergency situation, since elevated levels of solidarity and cooperation can often be observed during times of severe crisis. Therefore, the interesting question is: How long did this solidarity last? Are participants still in favor of the measures during the second lockdown and are they still complying with the measures or are there signs of fatigue?

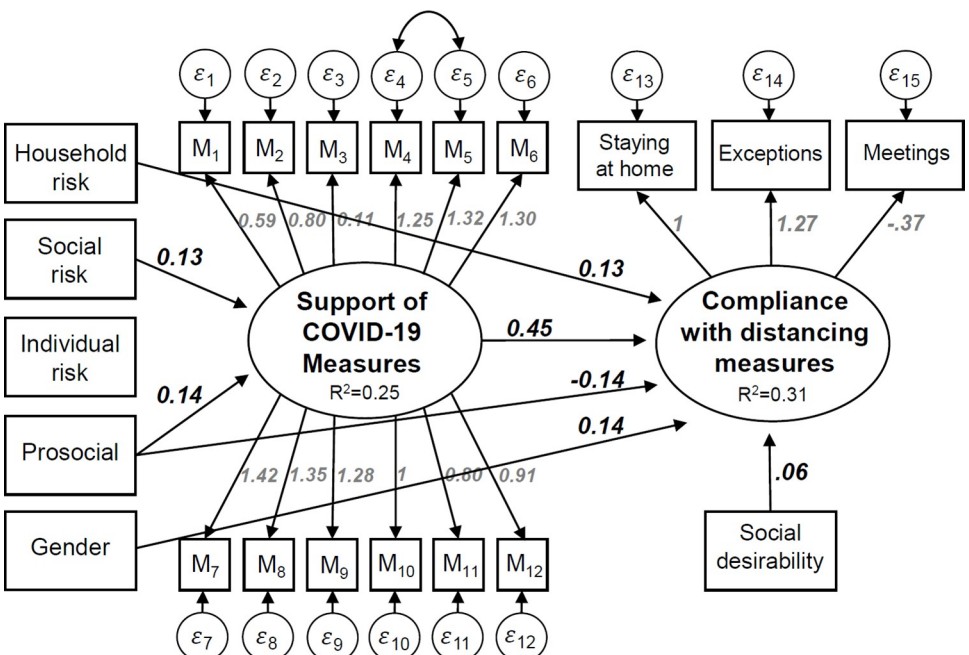

**Fig 1. Results of a structural equation model explaining compliance with Covid-19 measures during the first lockdown in Switzerland.** Note: N = 493. All reported coefficients are unstandardized and statistically significant at least at the 5% level [1].

Furthermore, based on the first wave, our first study showed that compliance predominately depends on support of the preventive measures, which in turn depends on social risk perception and pro-social orientation. This hypothesized causal structure is by and large confirmed by the structural equation model depicted in Fig 1 which we obtained from a student sample surveyed during the first lockdown in spring 2020 (see [1] for more details).

## 3. Methods and data

Our database for this paper is a two-wave panel study of a random sample of students of the University of Bern in Switzerland. Ethical clearance was obtained by the Ethics Committee of the Faculty of Business, Economics and Social Sciences of the University of Bern. The first wave of the survey was conducted at the end of the first lockdown in May 2020. For the second wave we recontacted the 510 participants of the first wave by the end of the second lockdown during April 2021.

As in the first wave, we conceptualize compliance with the social distancing measures as a latent construct and measure it using three indicators. First, on a 5-point Likert scale ranging from "not at all" (1) to "strictly" (5), participants of the second wave were asked to what extent they complied with the recommendation to stay at home as much as possible during the last four weeks before the interview (S1 Table in the supplement list the exact wording of all questions). The second indicator is the question of whether respondents made occasional exceptions to staying at home. The question has five answer categories ranging from "never" (1) to "very often" (5), and hence, is coded in the opposite direction as compared to the first indicator, balancing the index. For the statistical analysis, the coding of the answer categories of the second indicator is reversed. Thirdly, we asked how many friends or relatives participants met in their leisure time during the week before the interview. Meeting a large number of friends is not conforming to the coronavirus measures and hence indicates non-compliance to the safety

measures. For the analyses, we took the natural logarithm of the number of people, to achieve a more similar range and variance compared to the other two 5-point scales used for the construction of the latent compliance variable.

The main result of our first study is that compliance with the social distancing measures depends directly on the acceptance of the Covid-19 regulations. Individuals who agree with the regulations also show higher compliance with them (see Fig 1). We measure acceptance of the regulations again in the second wave by asking participants how much they support 11 different single measures implemented by the Swiss government during both lockdowns. For each measure respondents were asked to what extent they agree with the measure on a five-point answering scale ranging from "not at all" (1) to "very much" (5). The 11 single measures were: washing hands (M3), keeping social distance (M1), closure of restaurants and bars (M6), closure of universities (M5), not meeting more than 4 others (M10), closure of recreation facilities (M8), restrictions of public transportation (M11), closure of non-food stores (M7), border restrictions (M12), closure of schools (M4), and wearing of face masks (M2).

Support of the measures depends on individuals' risk perception regarding Covid-19. We differentiate between the risk participants perceive for themselves, members of the household, or for society in general. Individual risk perception is measured in both waves on 11-point scales asking participants how dangerous they believe a Covid-19 infection would be for themselves. Social risk is measured in both waves by how dangerous participants believe Covid-19 is for the health of the Swiss population. As it turned out in our first study, the perception that Covid-19 poses an individual risk is related to neither the acceptance of the measures nor to compliance with distancing rules. Possibly, this non-finding is due to the fact that we are dealing with a sample of young adults of whom many perceive a low personal risk. However, the perception that Covid-19 poses a health risk to others increased acceptance of the recommended regulations. Moreover, living together in a household with a person who has a health risk increased compliance with the coronavirus measures (see Fig 1).

The model also incorporates pro-sociality. Our analysis of wave 1 found that pro-socials expressed higher support for the measures [1] which in turn increases compliance; this is consistent with the findings of other studies [2, 10]. However, and surprisingly, we also found that the direct effect on compliance is negative. As in study 1, we measure pro-social orientation in the second wave via a revealed preference approach. Respondents received 10 Swiss Francs (about $10) for participation in the second wave. At the end of the questionnaire they had the option to donate some (or all) of their payment to a charitable organization of their choice. Those who donate some of their payments (about 52%) are classified as pro-socials. Furthermore, as in wave one we also include the Marlowe-Crowne scale, which measures social desirability in this second model [25, 26].

In wave two we also incorporate two new variables. Since former studies suggest that trust in the government and health institutions increases compliance, we also include trust in the government in this study. For this purpose, we asked participants to rate how much trust they have in politics on an 11-point scale ranging from 0 (no trust at all) to 10 (very much trust). Furthermore, former research shows that individual behavior is not only guided by injunctive norms, e.g. the regulations aimed at avoiding the spread of Covid-19, but also by descriptive norms (e.g. [27]). Therefore, we added a measure of the descriptive norm by asking participants to what extent their friends and acquaintances followed social distancing measures on a 5-point scale ranging from 1 (never) to 5 (always). The assumption is that compliance behavior is also guided by the social influence of what others do.

In the following section, we first present a few descriptive results of how the 364 participants changed in attitudes and behavior from the first to the second lockdown. In a second step, we re-estimate an extended version of the structural equation model depicted in Fig 1 to

see if the assumed causal structure can be replicated. Finally, we conduct a few checks to see if the causal structure assumed by the structural model is justified by using data from both waves.

## 4. Results

Of the 510 respondents of the first wave, 400 participated again in the second study. The online questionnaire of the second wave contained about 75 questions and the average completion time was 14 minutes. The online questionnaire contained an attention check by asking participants not to tick any of the given answers to a fake question which was placed in the middle of the questionnaire. 25 participants failed to pass this attention check by ticking an answer category and were excluded from further analysis. Moreover, we encountered 11 missing values in some of the variables, which leaves us with 364 valid cases. Descriptive information regarding all relevant variables of the second wave is presented in the supporting information in S2 Table. The mean age of the second wave is 24.2 years with a range of 19–36, and a female share of 64%. There are no statistically significant differences between the two waves with respect to age (t-value = 1.09, degrees of freedom = 855, two-sided p-value = 0.28) and the sex distribution (t-value = 0.27, degrees of freedom = 855, two-sided p-value = 0.79).

Fig 2 shows the comparison of the acceptance of the 11 most important anti Covid-19 measures between the first and second lockdowns. For descriptive purposes the two highest answer categories were collapsed into one agreement category. Hence, the bars denote the proportion of respondents who either accept or accept the measures very much. As can be seen the two measures of washing hands thoroughly and keeping social distance receive very high acceptance during both lockdowns.

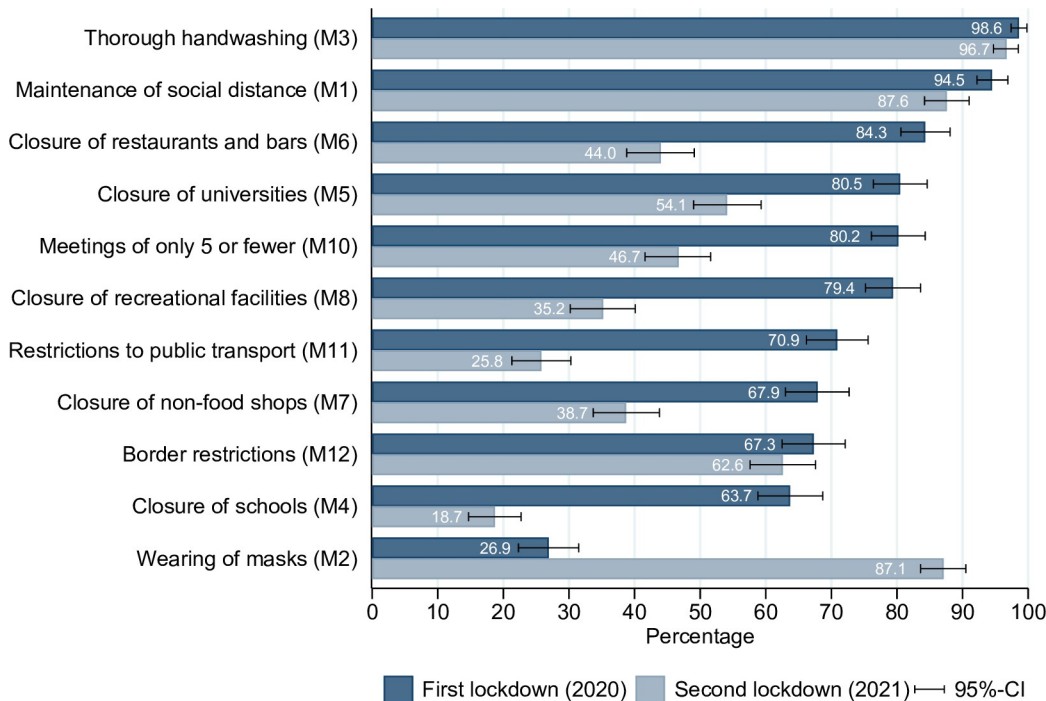

**Fig 2. Acceptance of Covid-19 measures during the first and second lockdowns.** Note: N = 364. Results for respondents who participated in both waves. Each measure was surveyed via five-point Likert scales ranging from "do not support at all" (1) to "support very much" (5). The figure displays the proportion of respondents supporting a measure weakly or strongly (categories 4 and 5).

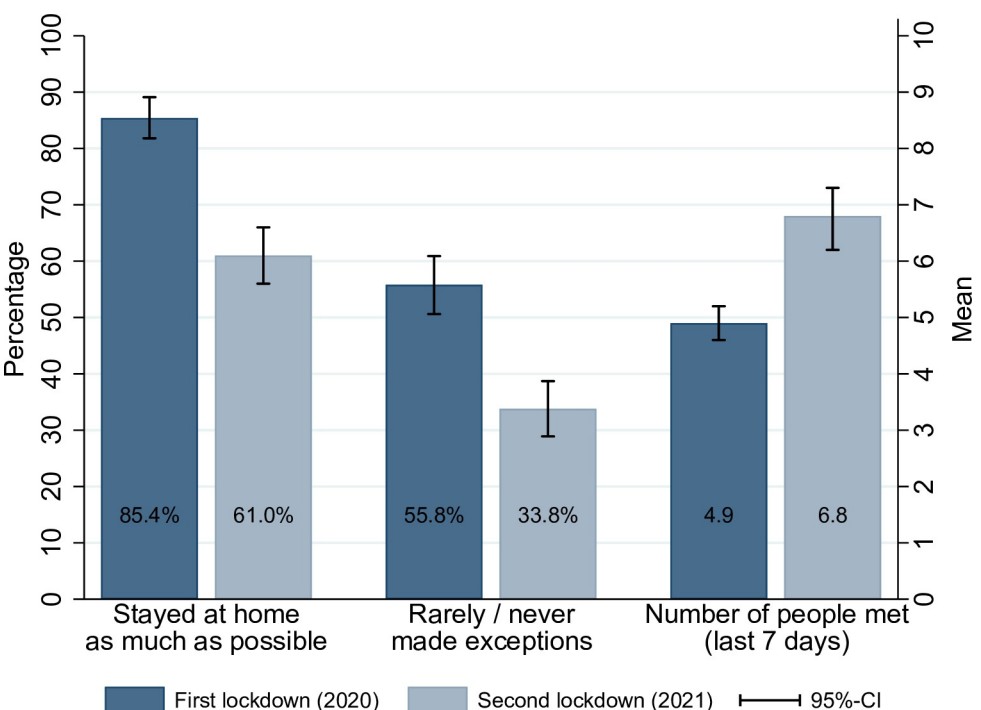

**Fig 3. Adherence to social distancing measures during first and second lockdowns.** Note: N = 364. Results for respondents who participated in both waves. Adherence was surveyed via five-point Likert scales, and the number of people met in the last 7 days. The figure displays the proportion of respondents, who stayed at home "mostly" or "strictly" (categories 4 and 5), the proportion of respondents, who "rarely" or "never" made exceptions (categories 4 and 5), and the average number of people met. The differences between the first and second wave are all statistically significant.

However, the acceptance of most other measures such as the closing of schools, universities, restaurants, recreational facilities, and restrictions of public transport dropped substantially by about half of the agreement rates during the first lockdown. One exception to this observation is the wearing of face masks, which increased substantially. This is most likely a reflection of the special situation in Switzerland, where the use of face masks was discouraged by health officials during the first lockdown. Later on officials changed their opinions on the preventive effects of face masks and so did participants in our study.

Next, Fig 3 depicts the change in social distancing behavior. For descriptive purposes the original categories "mostly" and "strictly" as well as "rarely" and "never" were collapsed into single categories, respectively. Hence, Fig 3 depicts the proportion of participants who adhere to the social distancing measures. As can be seen, adherence to social distancing decreased substantially during the second lockdown; from 85% who report staying at home as much as possible during the first lockdown to 61% (t-value 8.83, degrees of freedom = 363, two-sided p-value < 0.001) in the second lockdown, and from 55% who rarely or never made exceptions to 35% (t-value = 7.34, degrees of freedom = 363, two-sided p-value < 0.001). In line with these findings, the number of people respondents met during the week before the interview increased from an average of 4.9 to an average of 6.8 (t-value 7.01, degrees of freedom = 363, two-sided p-value < 0.001). As denoted in the brackets, all of these differences are statistically highly significant.

Next, we replicate the structural equation model which we calculated for the first lockdown [1]. Structural equation modeling allows the simultaneous estimation of latent constructs and of the structural relations between them (e.g. [28–30]). We estimate the model using Stata 17

[31–33]. We regress all exogenous variables on both latent constructs. The latent construct "Support of Covid-19 measures" includes 8 different preventive measures, and not 12 anymore as in the first model. We dropped "closure of parks" from the model, because this measure was not in practice anymore during the second lockdown. The three indicators M3, M11, and M12 have only low factor loadings, meaning that the Bentler-Raykov [34] squared multiple correlation coefficient is smaller than 0.2. Hence, we excluded them from the model. Furthermore, we include the error covariance between M1 (maintain social distance) and M2 (wearing face mask), M4 (closure of schools) and M5 (closure of universities), M6 (closure of restaurants) and M7 (closure of non-food shops), and M7 and M8 (closure of recreational facilities) as suggested by modification indices. Because our variables do not follow a multivariate normal distribution, we use maximum likelihood estimation applying the Satorra-Bentler correction for standard errors and model fit parameters [33, 35, 36]. The Satorra-Bentler corrected $\chi^2$ value of the model is 184.5 with 111 degrees of freedom. However, the $\chi^2$-statistic is not an ideal test statistic for this model, because of the large sample size and the relatively large number of indicators. Therefore, we follow the literature [37–39], and use CFI and TLI as goodness-of-fit statistics. The Satorra-Bentler corrected CFI equals 0.957 and the TLI goodness-of-fit statistic is 0.945. Both indicate a good model fit, since they reach the threshold of 0.95. The SRMR fit statistic has a value of 0.039, which is smaller than the threshold of 0.05, also suggesting a good model fit. Another test statistic is the RMSEA. In our case, RMSEA has a value of 0.044 with a 90% confidence interval (CI) of 0.033 to 0.055, which also indicates a good model fit. In addition, the test of close fit, testing that RMSEA is smaller than 0.05, is statistically not significant (p = 0.816), as desired. Altogether, the different goodness of fit statistics indicate that the model explains the data structure very well.

The unstandardized results of the model estimation for the second lockdown are shown in Fig 4. We regressed all exogenous variables on both latent variables. Effects that are statistically significant are depicted by arrows. Variables that are not connected by arrows with any of the two latent constructs did not show any statistically significant relation. By and large the model is a very good replication of the model constructed for the first lockdown. Most importantly, the latent variables "compliance with social distancing measures" and "support of Covid-19 measures" are again well measured by the three measures depicted in Fig 3, and respectively the 8 selected preventive measures (see Fig 2 and also supporting material in S2 Table). Furthermore, the main structural effects describing the influence on social distancing behavior are replicated. Hence, a positive attitude towards the coronavirus measures and living with a person at risk drives compliance behavior. Interestingly, we cannot replicate the effects of gender, pro-sociality and social desirability on compliance, which however had only a small impact in the first model.

Also, the structural effects of household risk, social risk, and prosocial attitudes on the acceptance of the measures, are replicated. There are also some new insights that emerge from this second model. First, political trust increases the acceptance of the Covid-19 measures, but it has no direct effect on compliance. Hence, the effect of political trust on compliance is mediated by attitudes towards the measures. Furthermore, the descriptive norm has a very strong effect on compliance of 0.56. Put differently, the perception that others comply is the strongest predictor of compliance in our model, i.e., an increase by one standard deviation in the descriptive norm increases compliance by 0.47 standard deviations. In comparison, an increase of one standard deviation of the attitudes towards the measures increases compliance by 0.29 standard deviations. Hence, attitudes still predict behavior, but the model depicted in Fig 4 suggests that the perception of what others do is more important. Overall, the model explains 48% of the variance of the social distancing compliance, and 50% of the support of the preventive measures.

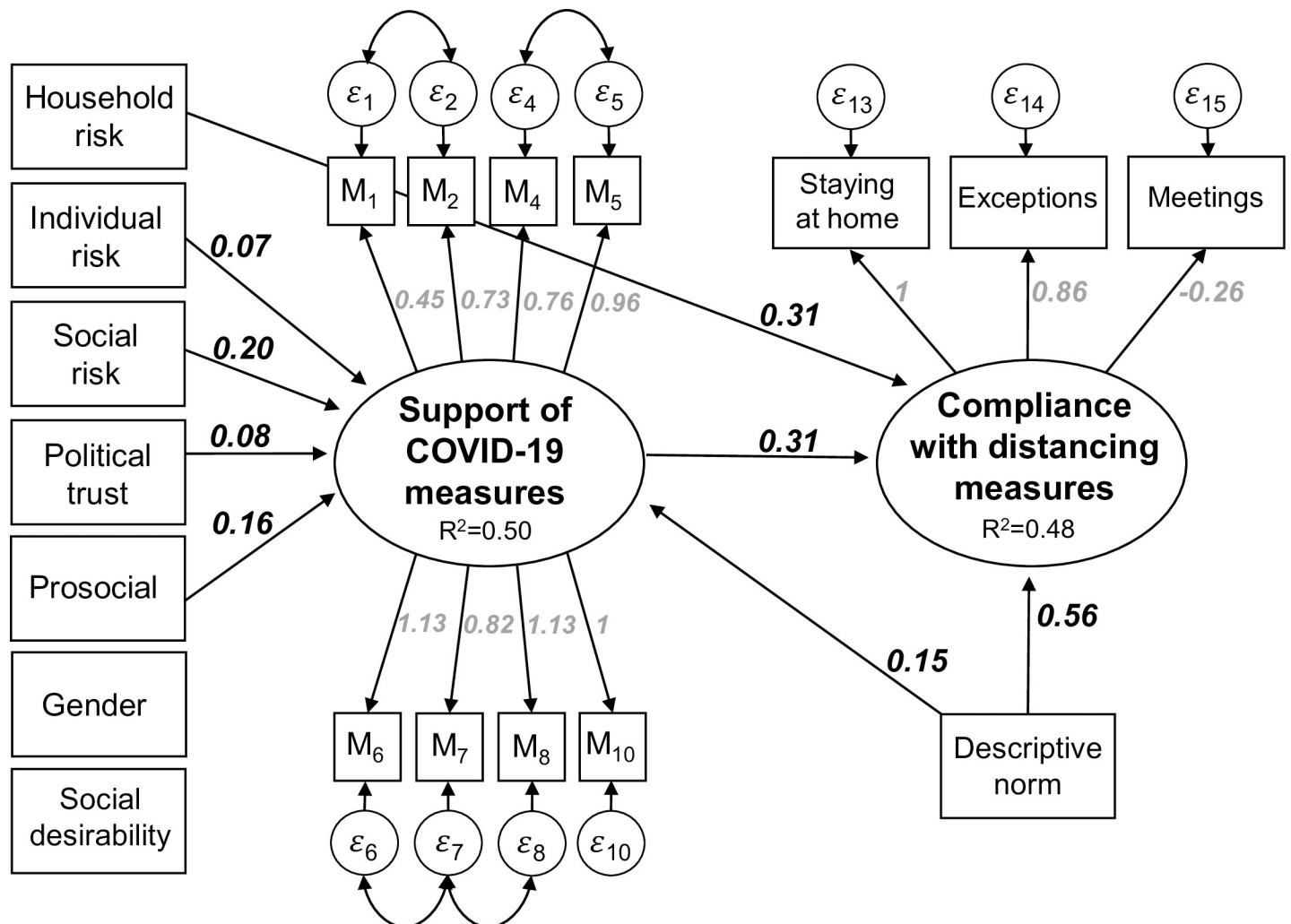

**Fig 4. A structural equation model of compliance with the coronavirus distancing measures.** Note: N = 364. All reported coefficients are unstandardized and statistically significant at least at the 5%-level.

As with any attitude-behavior model that is solely tested via cross-sectional data analysis, we cannot exclude the possibility that favorable attitudes of the coronavirus measures are not the cause of compliance behavior but simply a rationalization of it. Hence, from the evidence presented so far it is theoretically possible that causality runs the other way and that behavior causes attitudes. A stronger test of causality is possible by using panel data. In what follows we conduct two more stringent tests by utilizing the panel structure of the data. If compliance behavior causes attitudes, then the compliance behavior participants reported in 2020 during the first lockdown should influence their attitudes measured in 2021 during the second lockdown. We tested this assumption by incorporating compliance behavior measured in 2020 into the structural equation model presented in Fig 4. We used the eight indicators of support of the Covid-19 measures in 2021 (M1, M2, M4, M5, M6, M7, M8, M10), and we measured compliance with distancing measures in 2020 and 2021 respectively with the three indicators each, which can be found in Fig 3. Modification indices suggested the addition of an error covariance between the number of people met in 2020 and in 2021. Theoretically, it makes sense to include it for two reasons: First, because some people meet more people in general

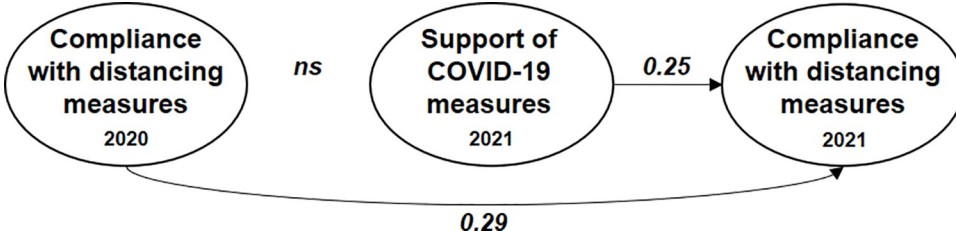

**Fig 5. A test of the assumed causal structure via structural equation modeling.** Note: N = 364. All reported coefficients are standardized and statistically significant at least at the 5%-level. Included control variables are individual risk, social risk, household risk, descriptive norm, trust in politics, prosocial, gender, and social desirability.

than others. Second, the scale is different from the two other indicators used when measuring compliance. Also, we included the covariance between making exceptions of wave one and wave two. This model fits the data very well ($\chi^2$ = 257, df = 156, CFI = 0.956, TLI = 0.942, SRMR = 0.042, RMSEA = 0.042, RMSEA 90%-CI = [0.033, 0.051]). The extended model explains 54% of the variance in the social distancing compliance, and 50% of the support of the Covid-19 measures.

The standardized results of this test are shown in Fig 5. For simplicity, we present only the structural coefficients of interest. However, all other results remain very similar as shown in Fig 4. As can be seen, the analysis does not result in a statistically significant effect of behavior in 2020 on attitudes in 2021. However, the social distancing behavior shown by participants in 2020 does influence their behavior in 2021. Hence, those who complied with the coronavirus measures in 2020 also did so in 2021, independently of their attitudes in 2021. However, in addition to the effect of the past behavior the effect of attitudes on behavior remains significant. Inclusion of the past behavior decreases the influence of attitudes, but an increase of one standard deviation of attitudes still increases compliance behavior by 0.25 standard deviations. Hence, attitudes still have a strong influence on compliance behavior even if past behavior is taken into account. This result confirms the assumed causal structure of the model depicted in Fig 4.

Another test of causality can be obtained by estimating a fixed effects panel regression. Hence, we are regressing the difference in behavior between the first and the second lockdown on the difference in attitudes between the lockdowns together with other time-varying variables. The structural equation model already indicates that three variables have direct effects on compliance. Next to attitudes these are the variables of household risk, and the descriptive norm. Since the descriptive norm was only measured in the second wave, the variable cannot be included in a first difference model. Therefore, our model contains three variables; the difference in attitudes towards the measures, the changes in household risk, and a time dummy variable indicating the second wave. The model can be written as follows:

$$Y_{it} - Y_{it-1} = \beta_0 + \beta_1(X_{1it} - X_{1it-1}) + \beta_2(X_{2it} - X_{2it-1}) + \beta_3 T + (\varepsilon_1 - \varepsilon_2)$$

The first term describes the difference of behavior of individual $i$ between the two measurements. The difference in attitudes of individual $i$ between the two measurements is denoted by $X_{i1t}$-$X_{i1t-1}$, and the change in household risk is denoted by $X_{i2t}$-$X_{i2t-1}$. T is a dummy variable denoting the second wave and catches all differences not otherwise accounted for that might occur between the two time periods. Finally, epsilon 1 and 2 denote the error terms of the two waves. The advantage of such a two-way fixed effects model is that it takes only the within individual variance into account for the estimation of the coefficients, and not the between individual variances that might be biased due to unobserved heterogeneity. Two-way fixed effects

models are seen in the literature as the best way to estimate unbiased causal effects of an independent variable X on Y [40, 41]. The estimation can of course still suffer from bias if there are measurement errors in any of the dependent or independent variables. Also, reversed causality can still be a problem in fixed effects regressions. However, we showed already in Fig 5 that reversed causality is very unlikely in our case.

To estimate a fixed effect panel regression the former latent variables need to be transformed into manifest variables. We do this by constructing indices. For the compliance index, the three indicators were summed up to an index. Hence, individuals who strictly stayed at home (ranging from 0 to 4), never made exceptions (ranging from 0 to 4), and who met no friends during the week before the interview receive the highest value on the index. For the purpose of constructing the index, the latter variable (meeting friends) was reversely coded by subtracting the maximum value of the log number of friends met (ranging from 0 to 3.87 in 2021, resp. to 3.04 in 2020).

For 2021, the index ranges from 0.61 to 11.87, and has a decent reliability in terms of Cronbach's alpha of 0.74. For 2020, the index ranges from 1.21 to 11.04, and has a Cronbach's alpha of 0.65. For the construction of the index of the attitudes towards the measures we took the 8 items (M1, M2, M4, M5, M6, M7, M8, M10) and summed the values of each item. Hence, the index runs from 8 (low approval of the measures) to 40 (high approval of the measures). The Cronbach's alpha of this scale is 0.85 for 2021 and 0.83 for 2020 indicating very high reliability. Descriptive information regarding the four indices is presented in the supporting information in S3 Table. The results of the fixed effects regression are presented in Table 1. As can be seen, the change in attitudes towards the measure is statistically significantly related to compliance behavior. An increase of one standard deviation in attitudes increases compliance by 0.34 standard deviations. Hence, the effect is similar in size as already obtained in the structural equation model. A change in household risk is not reliably related to a change in compliance

**Table 1. Two-way fixed effects panel regression on the change of complying with social distancing measures.**

| Variables | Coefficient |
|---|---|
| Support of COVID-19 measures | 0.34*** |
| | (0.059) |
| Household risk | |
| Transition to household with person at risk[§] | 0.35 |
| | (0.183) |
| Leaving of household with person at risk[§] | 0.07 |
| | (0.186) |
| Second Wave[§] | -0.01 |
| | (0.073) |
| Constant | -0.01 |
| | (0.046) |
| Within $R^2$ | 0.143 |
| n | 364 |
| n x T | 728 |

Note
* = $p < 0.05$
** = $p < 0.01$
*** = $p < 0.001$. Standardized regression coefficients with standard errors in brackets.
§ = non-standardized dichotomous variables.

behavior, and also the time dummy variable is not statistically significantly related to compliance indicating that there are no unaccounted further effects present.

## 5. Summary and discussion

The analysis of two-wave panel data collected during the first and second Covid-19-induced lockdowns reveals a few interesting results. First of all, we observe a fatigue effect in our sample of young adults. While the coronavirus measures received very high support during the first lockdown, and while study participants report high compliance during this period, support of the measures and compliance with social distancing decreased considerably during the second lockdown. This result is in line with other studies, e.g. for the UK [42] which reports that compliance decreased significantly during a period of 5 month. Second, a structural equation model shows that social distancing compliance is most importantly driven by the descriptive norm and the acceptance of the Covid-19 measures. Hence, respondents who believe that others are adhering to the social distancing rules also keep the rules themselves. Furthermore, our results suggest that many effects on compliance reported in the literature are mediated by the attitudes towards the measures. Thus, individual and social risk perception, trust in politics, and prosocial attitudes increase the acceptance of the measures which in turn leads to stronger adherence to the social distancing measures. Moreover, we used the data of both waves to conduct further tests on the causality of the relationship between attitudes and behavior. Both test results suggest that attitudes drive the compliance behavior and that the obtained relation can be interpreted causally.

While these results are interesting and add new insights to the existing literature on attitudes and compliance behavior during the Covid-19 pandemic, our study also has some limitations. Most importantly, we do not have a random population sample but only a random sample of students from the University of Bern. Hence, our sample consists of young and educated adults, and the results cannot be generalized to the whole population or to the whole younger generation. Particularly, the homogeneity with respect to age and education makes it impossible to assess how these socio-demographic characteristics influence compliance. The study is of course also limited to one region within Switzerland. Since other countries reacted with much more restrictive measures (e.g. including curfew) to the spread of Covid-19 or penalized non-compliant behavior more heavily, inhabitants of these countries might have responded differently in terms of acceptance of the measures as well as in terms of complying with the measures. Hence, international comparisons of how the different measures affected attitudes and compliance would certainly enrich the insights of Covid-19 related attitudes and behaviors.

## Supporting information

**S1 Table. List of variables and question wording.**
(DOCX)

**S2 Table. Descriptive information of variables of wave two (2021).**
(DOCX)

**S3 Table. List of index variables for the first wave (2020), and the second wave (2021).**
(DOCX)

## Author Contributions

**Conceptualization:** Axel Franzen.

**Data curation:** Fabienne Wöhner.

**Formal analysis:** Axel Franzen, Fabienne Wöhner.

**Investigation:** Axel Franzen, Fabienne Wöhner.

**Methodology:** Axel Franzen, Fabienne Wöhner.

**Project administration:** Axel Franzen.

**Supervision:** Axel Franzen.

**Visualization:** Fabienne Wöhner.

**Writing – original draft:** Axel Franzen.

**Writing – review & editing:** Axel Franzen, Fabienne Wöhner.

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
