## [Decision Letter · Decision Letter 0]

28 Sep 2021

PONE-D-21-25762Fatigue during the COVID-19 pandemic: Evidence of social distancing adherence from a panel study of young adults in SwitzerlandPLOS ONE

Dear Dr. Franzen,

Thank you for submitting your manuscript to PLOS ONE. After careful consideration, we feel that it has merit but does not fully meet PLOS ONE’s publication criteria as it currently stands. Therefore, we invite you to submit a revised version of the manuscript that addresses the points raised during the review process.

I concur with the major revisions proposed by both reviewers. In particular, the issues pertaining to construct validity and robustness of the structural equation model need to be thoroughly addressed in the revised version. Comparison of your findings with other similar studies can also be enhanced in the discussion section.

We look forward to receiving your revised manuscript.

Kind regards,

Asim Zia, Ph.D.

Academic Editor

PLOS ONE

Additional Editor Comments:

I concur with the major revisions proposed by both reviewers. In particular, the issues pertaining to construct validity and robustness of the structural equation model need to be thoroughly addressed in the revised version. Comparison of your findings with other similar studies can also be enhanced in the discussion section.

Reviewers' comments:

Reviewer's Responses to Questions

**Comments to the Author**

1. Is the manuscript technically sound, and do the data support the conclusions?

Reviewer #1: Partly

Reviewer #2: No

2. Has the statistical analysis been performed appropriately and rigorously? 

Reviewer #1: Yes

Reviewer #2: Yes

3. Have the authors made all data underlying the findings in their manuscript fully available?

Reviewer #1: Yes

Reviewer #2: Yes

4. Is the manuscript presented in an intelligible fashion and written in standard English?

Reviewer #1: Yes

Reviewer #2: Yes

5. Review Comments to the Author

Reviewer #1: This study capitalizes on an interesting feature of the COVID-19 pandemic in Switzerland to help understand changes in compliance behaviors in young adults. The manuscript has informative figures and provides new, and thought-provoking information related to distancing behaviors, norms, and pandemic fatigue during COVID-19. The sample and dataset are unique and I expect will be of interest to the field. Some errors and oversights reduce confidence in results. My comments are listed below:

1. Is it possible that something other than “pandemic fatigue” could explain the reduction of these compliance behaviors between 2020-2021? For example, it seems clear that face masks are more widely used/available at the time of the 2021 data collection based on the change in acceptance rate. So, leaving home and meeting with friends would be fundamentally less risky (if using a face mask) than before. Can the authors address this alternative? Relatedly, in the panel regression -- does removing the face mask item from the support of COVID-19 measures variable alter results? Should that particular item be included as a separate predictor since it is in the opposite direction of all other support items and therefore makes interpretation difficult?

2. The meaning of all M1-M12 variables should be included somewhere in the main text. The authors also do not describe what the “Social desirability” variable is in this paper.

3. The discussion section does not adequately relate current findings to the literature. Alternative explanations should also be described.

4. In this second wave of data collection, the compliance question specified participants to respond about their behavior in the past 4-weeks. Can the authors clarify whether this differs from the first wave and, if so, whether this presents any limitation?

5. On line 368, the authors state “All of these differences are statistically highly significant”, but results of statistical tests are not provided.

6. For the Table 1 legend, it may be helpful to state that the dependent variable of the model is the difference in compliance behavior.

7. Table S2 has a typo that shows 2021 compliance twice.

8. I believe there is a rounding typo related to the number of participants reporting staying at home as much as possible in 2021 (Figure 3/ line 365).

9. Authors should standardize rounding where appropriate (e.g., on page 13) and check for typographical/ grammar errors throughout, including on lines: 208, 253, 286, 499, 552.

Reviewer #2: This paper examines predictors of support for COVID-19 measures and compliance with social distancing in a panel of young adults in Switzerland. The data and results are interesting and promising. However, the paper could benefit from revisions clarifying the survey instruments and construction of variables and SEM model structures, including describing models in a way that makes clear what associations were tested but found to be statistically insignificant.

Introduction:

• I don’t believe that any papers from the University of Southern California’s Understanding America Survey (UAS) were included in the literature review. That is a national US panel of United States adults which conducted a longitudinal survey of COVID-19 beliefs and behaviors. For example, Relationships between initial COVID-19 risk perceptions and protective health behaviors: a national survey WB de Bruin, D Bennett - American Journal of Preventive Medicine, 2020 would be worth citing in the literature review

Methods:

• I believe that the first mention that the study sample was recruited from University of Bern is in the methods. Prior descriptions in the abstract and introduction led me to assume that recruitment was a broader sample of young adults in Switzerland. I recommend adding a sentence to both the abstract and introduction describing the recruitment sample.

• I would suggest moving the description of the wave 2 sample to the results section (Starting with the sentence on line 258 starting with “Of these, 400 participated…”. The description of the attention check question should remain in the methods, but the results describing how many participants were excluded due to that question should be moved to results.

• I think that the clarity of the methods section could be improved by clearly describing in separate subsections: (1) the survey instrument, (2) how the survey responses were processed into variables for the statistical model, (3) the analyses that will be presented in the results.

• The exact phrasing (or a translation) of the survey instruments should be included somewhere. (Can be in the supplement). If the exact phrasing was described in the prior study for most instruments, that should be clearly stated and the phrasing of the new questions should be included.

Results:

• As mentioned above, the overview of the analyses should be moved to the methods.

• Figure 2 would benefit from inclusion of error bars similar to figure 3.

• This paper: Hu, Li‐tze, and Peter M. Bentler. "Cutoff criteria for fit indexes in covariance structure analysis: Conventional criteria versus new alternatives." Structural equation modeling: a multidisciplinary journal 6.1 (1999): 1-55. Suggests that cutoff values for CFI and TLI to assess fit should be 0.95 rather than 0.9. At a minimum, the authors should examine the residual covariance matrix and report the results of that analysis.

• As the results are currently displayed for the SEMs, it is not easy to understand which relationships were tested but not found to be significant. It would be helpful to depict somewhere – either in the same SEM results diagrams or elsewhere – relationships included in the model but found to be statistically insignificant.

• A more minor point is that Figure 1 appears to include results for the full sample that the authors previously published. It would be a cleaner comparison to re-do that same analysis for the subset of respondents who responded to the second survey and included in that analysis.

• It is not exactly clear how all of the variables in the regression model were constructed. For example, the “meetings” compliance indicator had a negative association with the latent variable in the SEMs. Was that measure reverse coded when used to construct Y_it?

6. PLOS authors have the option to publish the peer review history of their article (what does this mean?). If published, this will include your full peer review and any attached files.

Reviewer #1: No

Reviewer #2: No

---

## [Author Response · Author response to Decision Letter 0]

13 Oct 2021

Dear Asim Zia

First of all, we thank you and the two reviewers very much for the effort of reviewing our manuscript and for the helpful suggestions. We incorporated almost all of them and respond to every comment point by point as they were made by the reviewers. 

Response to reviewer 1: 

1) The possibility that people met more often in 2021 as compared to 2020 because masks were viewed as preventing infections is a good comment. However, risk perception did not change between 2020 and 2021 which speaks against this possibility. Furthermore, we recalculated the panel regression model, once with the index excluding wearing face masks, and second including additionally the face masks as a separate indicator. First, results of index without face masks does not change the results. Second, treating the face masks as an additional indictor has no statistically significant effect. The table with the regression is shown below. Hence, face masks have no influence on the compliance with social distancing and there is no need for any change of the manuscript. 

Please find Table 1 in the "response to reviewers" document.

2) Thank you for the comment. We include the abbreviations M1 through M12 to the description of the measures on page 9 and in Figure 2. Furthermore, we include a sentence that we measure social desirability by the Marlowe-Crowne scale on page 10 and provide two new references. 

3) Good comment, but please note that our study is one of the first that reports panel results comparing different lock down periods. However, we did find one study that fits into the discussion part and we refer to it in the revised version. 

4) Yes, in both waves the questions referring to how much respondents stayed at home refers to a four-week period. 

5) Thank you for the comment. We included the results of the significance tests in the revised version. 

6) Thank you for the comment. We rephrased the title of Table 1 to “Two-way fixed effects panel regression on the change of complying with social distancing measures”. 

7) Thank you for reading also the supplement. We corrected the typo. 

8) We checked the numbers again. It is exactly 60.98 and we rounded to 61.0% in Fig 3. We also corrected it to 61% in the text. 

9) Thank you very much for pointing out the typos which we corrected. 

Response to reviewer 2: 

1) Yes, thank you for the comment, we included the paper by Wändi Bruine de Bruin and Daniel Bennett (2020). 

2) We added in the abstract that we have a sample of young adults from the University of Bern, Switzerland. 

3) Thank you for the comment: As suggested, we removed the mentioned part from the Methods section into the Results section. 

4) Thank you for the comment. We think that clarity improved by following suggestion 3. The rest of the method section describes the survey instruments (the measurement of variables), and how the variables were coded step by step. We think this is easier to comprehend instead of first describing all the questions and then the coding afterwards. We believe this way we avoid unnecessary redundancies. Similar arguments apply to the description of the statistical procedures. Since we have more than one statistical model (structural equation model and panel regression), we believe describing first each model followed by its results is easier to comprehend as compared to describing first all models and afterwards all results. 

5) Thank you for this suggestion. We include the exact question wording in a new table (S1 Table) in the supporting information and refer to it in the text.

6) We followed this suggestion and moved the overview of the analyses into the method section. 

7) Thank you for this suggestion. We incorporated the error bars into Fig 2. 

8) This is a very helpful comment. We recalculated the model dropping three of the eleven indicators of the acceptance of the COVID-19 measures. These indicators were described as weak indicators already in the first version of the manuscript. Dropping the three indicators improved the model fit CFI and TLI to 0.957 and 0.945 respectively. None of the structural estimates was affected by this slight change of the measurement of the latent construct “support of COVID-19 measures”. We rewrote the results section accordingly, which makes it also more comprehensible. 

9) We also included a few sentences explaining which relations had been tested but not found significant. The new sentences are marked in the revised version. 

10) Thank for this comment. It is a good point and we considered it thoroughly. However, the model in Figure 1 is a reference of the results of the first wave. A replication of it with the data of wave 2 is presented in Figure 4. The results are very similar. Hence, it would be redundant to present the model with the data of wave 2 twice with only minor differences. Therefore, we believe it is better to leave Fig 1 as is, and we hope that the reviewer can accept our argument. 

11) We did included a couple of sentence improving the explanation of the coding and construction of the index of the dependent variable of the panel regression model.

---

## [Editor Report · Decision Letter 1]

29 Nov 2021

Fatigue during the COVID-19 pandemic: Evidence of social distancing adherence from a panel study of young adults in Switzerland

PONE-D-21-25762R1

Dear Dr. Franzen,

We’re pleased to inform you that your manuscript has been judged scientifically suitable for publication and will be formally accepted for publication once it meets all outstanding technical requirements.

Kind regards,

Asim Zia, Ph.D.

Academic Editor

PLOS ONE

Additional Editor Comments (optional):

All the issues raised by the reviewers in the first round have been successfully addressed. Very timely contribution!
---

## [Editor Report · Acceptance letter]

2 Dec 2021

PONE-D-21-25762R1 

Fatigue during the COVID-19 pandemic: Evidence of social distancing adherence from a panel study of young adults in Switzerland 

Dear Dr. Franzen:

I'm pleased to inform you that your manuscript has been deemed suitable for publication in PLOS ONE. Congratulations! Your manuscript is now with our production department. 

Kind regards, 

on behalf of

Professor Asim Zia 

Academic Editor

PLOS ONE